# NADS: Neural Architecture Distribution Search for Uncertainty Awareness

## Abstract

Machine learning systems often encounter Out-of-Distribution (OoD) errors when dealing with testing data coming from a different distribution from the one used for training. With their growing use in critical applications, it becomes important to develop systems that are able to accurately quantify its predictive uncertainty and screen out these anomalous inputs. However, unlike standard learning tasks, there is currently no well established guiding principle for designing architectures that can accurately quantify uncertainty. Moreover, commonly used OoD detection approaches are prone to errors and even sometimes assign higher likelihoods to OoD samples. To address these problems, we first seek to identify guiding principles for designing uncertainty-aware architectures, by proposing *Neural Architecture Distribution Search* (NADS). Unlike standard neural architecture search methods which seek for a single best performing architecture, NADS searches for a distribution of architectures that perform well on a given task, allowing us to identify building blocks common among all uncertainty aware architectures. With this formulation, we are able to optimize a stochastic outlier detection objective and construct an ensemble of models to perform OoD detection. We perform multiple OoD detection experiments and observe that our NADS performs favorably compared to state-of-the-art OoD detection methods.

## 1 Introduction

Detecting anomalous data is crucial for safely applying machine learning in autonomous systems for critical applications and for AI safety (Amodei et al., 2016). Such anomalous data can come in settings such as in autonomous driving (Kendall & Gal, 2017; NHTSA, 2017), disease monitoring (Hendrycks & Gimpel, 2016), and fault detection (Hendrycks et al., 2019b). In these situations, it is important for these systems to reliably detect abnormal inputs so that their occurrence can be overseen by a human, or the system can proceed using a more conservative policy.

The widespread use of deep learning models within these autonomous systems have aggravated this issue. Despite having high performance in many predictive tasks, deep networks tend to give high confidence predictions on Out-of-Distribution (OoD) data (Goodfellow et al., 2015; Nguyen et al., 2015). Moreover, commonly used OoD detection approaches are prone to errors and even assign higher likelihoods to samples from other datasets (Lee et al., 2018; Hendrycks & Gimpel, 2016).

Unlike common machine learning tasks such as image classification, segmentation, and speech recognition, there are currently no well established guidelines for designing architectures that can accurately screen out OoD data and quantify its uncertainty. Such a gap in our knowledge makes Neural Architecture Search (NAS) a promising option to explore the better design of uncertainty-aware models (Elsken et al., 2018). NAS algorithms attempt to find an optimal neural network architecture for a specific task. Existing efforts have primarily focused on searching for architectures that perform well on image classification or segmentation. However, it is unclear whether architecture components that are beneficial for image classification and segmentation models would also lead to better uncertainty quantification and thereafter be effective for OoD detection. Moreover, previous work on deep uncertainty quantification shows that ensembles can help calibrate OoD classifier based methods, as well as improve OoD detection performance of likelihood estimation models (Lakshminarayanan et al., 2017; Choi & Jang, 2018). Because of this, instead of a single

best performing architecture for uncertainty awareness, one might consider a distribution of well-performing architectures.

Along this direction, designing an optimization objective which leads to uncertainty-aware models is also not straightforward. With no access to labels, unsupervised/self-supervised generative models which maximize the likelihood of in-distribution data become the primary tools for uncertainty quantification (Hendrycks et al., 2019a). However, these models counter-intuitively assign high likelihoods to OoD data (Nalisnick et al., 2019a; Choi & Jang, 2018; Hendrycks et al., 2019a; Shafaei et al.). Because of this, maximizing the log-likelihood is inadequate for OoD detection. On the other hand, Choi & Jang (2018) proposed using the Widely Applicable Information Criterion (WAIC) (Watanabe, 2013), a penalized log-likelihood score, as the OoD detection criterion. However, the score was approximated using an ensemble of models that was trained on maximizing the likelihood and did not directly optimize the WAIC score.

To this end, we propose a novel *Neural Architecture Distribution Search* (**NADS**) framework to identify common building blocks that naturally incorporate model uncertainty quantification and compose good OoD detection models. NADS is an architecture search method designed to search for a distribution of well-performing architectures, instead of a single best architecture by formulating the architecture search problem as a stochastic optimization problem. Using NADS, we optimize the WAIC score of the architecture distribution, a score that was shown to be robust towards model uncertainty. Such an optimization problem with a stochastic objective over a probability distribution of architectures is unamenable to traditional NAS optimization strategies. We make this optimization problem tractable by taking advantage of weight sharing between different architectures, as well as through a parameterization of the architecture distribution, which allows for a continuous relaxation of the discrete search problem. Using the learned posterior architecture distribution, we construct a Bayesian ensemble of deep models to perform OoD detection. Finally, we perform multiple OoD detection experiments to show the efficacy of our proposed method.

## 2 BACKGROUND

### 2.1 NEURAL ARCHITECTURE SEARCH

Neural Architecture Search (NAS) algorithms aim to automatically discover an optimal neural network architecture instead of using a hand-crafted one for a specific task. Previous work on NAS has achieved successes in image classification (Pham et al., 2018), image segmentation (Liu et al., 2019), object detection (Ghiasi et al., 2019), structured prediction (Chen et al., 2018), and generative adversarial networks (Gong et al., 2019). However, there has been no NAS algorithm developed for uncertainty quantificaton and OoD detection.

NAS consists of three components: the proxy task, the search space, and the optimization algorithm. Prior work in specifying the search space either searches for an entire architecture directly, or searches for small cells and arrange them in a pre-defined way. Optimization algorithms that have been used for NAS include reinforcement learning (Baker et al., 2017; Zoph et al., 2018; Zhong et al., 2018; Zoph & Le, 2016), Bayesian optimization (Jin et al., 2018), random search (Chen et al., 2018), Monte Carlo tree search (Negrinho & Gordon, 2017), and gradient-based optimization methods (Liu et al., 2018b; Ahmed & Torresani, 2018). To efficiently evaluate the performance of discovered architectures and guide the search, the design of the proxy task is critical. Existing proxy tasks include leveraging shared parameters (Pham et al., 2018), predicting performance using a surrogate model (Liu et al., 2018a), and early stopping (Zoph et al., 2018; Chen et al., 2018).

To our best knowledge, all existing NAS algorithms seek a single best performing architecture. In comparison, searching for a distribution of architectures allows us to analyze the common building blocks that all of the candidate architectures have. Moreover, this technique can also complement ensemble methods by creating a more diverse set of models for the ensemble decision, an important ingredient for deep uncertainty quantification (Lakshminarayanan et al., 2017).

### 2.2 UNCERTAINTY QUANTIFICATION AND OUT-OF-DISTRIBUTION DETECTION

Prior work on uncertainty quantification and OoD detection for deep models can be divided into model-dependent (Lakshminarayanan et al., 2017; Gal & Ghahramani, 2016; Liang et al., 2017),

and model-independent techniques (Dinh et al., 2016; Germain et al., 2015; Oord et al., 2016). Model-dependent techniques aim to yield confidence measures $p(y|\boldsymbol{x})$ for a model's prediction $y$ when given input data $\boldsymbol{x}$. However, a limitation of model-dependent OoD detection is that they may discard information regarding the data distribution $p(\boldsymbol{x})$ when learning the task specific model $p(y|\boldsymbol{x})$. This could happen when certain features of the data are irrelevant for the predictive task, causing information loss regarding the data distribution $p(\boldsymbol{x})$. Moreover, existing methods to calibrate model uncertainty estimates assume access to OoD data during training (Lee et al., 2018; Hendrycks et al., 2019b). Although the OoD data may not come from the testing distribution, this assumes that the structure of OoD data is known ahead of time, which can be incorrect in settings such as active/online learning where new training distributions are regularly encountered.

On the other hand, model-independent techniques seek to estimate the likelihood of the data distribution $p(\boldsymbol{x})$. These techniques include Variational Autoencoders (VAEs) (Kingma & Welling, 2013), generative adversarial networks (GANs) (Goodfellow et al., 2014), autoregressive models (Germain et al., 2015; Oord et al., 2016), and invertible flow-based models (Dinh et al., 2016; Kingma & Dhariwal, 2018). Among these techniques, invertible models offer exact computation of the data likelihood, making them attractive for likelihood estimation. Moreover, they do not require OoD samples during training, making them applicable to any OoD detection scenario. Thus in this paper, we focus on searching for invertible flow-based architectures, though the presented techniques are also applicable to other likelihood estimation models.

Along this direction, recent work has discovered that likelihood-based models can assign higher likelihoods to OoD data compared to in-distribution data (Nalisnick et al., 2019a; Choi & Jang, 2018) (see Figure 13 for an example). One hypothesis for such a phenomenon is that most data points lie within the typical set of a distribution, instead of the region of high likelihood (Nalisnick et al., 2019b). Thus, Nalisnick et al. (2019b) recommend to estimate the entropy using multiple data samples to screen out OoD data instead of using the likelihood. Other uncertainty quantification formulations can also be related to entropy estimation (Choi & Jang, 2018; Lakshminarayanan et al., 2017). However, it is not always realistic to test multiple data points in practical data streams, as testing data often come one sample at a time and are never well-organized into in-distribution or out-of-distribution groups.

With this in mind, model ensembling becomes a natural consideration to formulate entropy estimation. Instead of averaging the entropy over multiple data points, model ensembles produce multiple estimates of the data likelihood, thus "augmenting" one data point into as many data points as needed to reliably estimate the entropy. However, care must be taken to ensure that the model ensemble produces likelihood estimates that agree with one another on in-distribution data, while also being diverse enough to discriminate OoD data likelihoods. In what follows, we propose NADS as a method that can identify distributions of architectures for uncertainty quantification. Using a loss function that accounts for the diversity of architectures within the distribution, NADS allows us to construct an ensemble of models that can reliably detect OoD data.

## 3 NEURAL ARCHITECTURE DISTRIBUTION SEARCH (NADS)

Putting Neural Architecture Distribution Search (NADS) under a common NAS framework (Elsken et al., 2018), we break down our search formulation into three main components: the proxy task, the search space, and the optimization method. Specifying these components for NADS with the ultimate goal of uncertainty quantification for OoD detection is not immediately obvious. For example, naively using data likelihood maximization as a proxy task would run into the issue pointed out by Nalisnick et al. (2019a), with models assigning higher likelihoods to OoD data. On the other hand, the search space needs to be large enough to include a diverse range of architectures, yet still allowing a search algorithm to traverse it efficiently. In the following sections, we motivate our decision on these three choices and describe these components for NADS in detail.

### 3.1 PROXY TASK

The first component of NADS is the training objective that guides the neural architecture search. Different from existing NAS methods, our aim is to derive an ensemble of deep models to improve model uncertainty quantification and OoD detection. To this end, instead of searching for

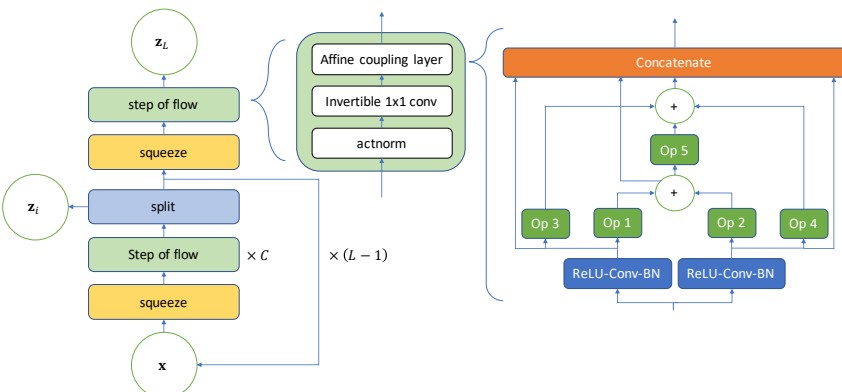

Figure 1: Search space of a single block in the architecture

architectures which maximize the likelihood of in-distribution data, which may cause our model to incorrectly assign high likelihoods to OoD data, we instead seek architectures that can perform entropy estimation by maximizing the Widely Applicable Information Criteria (WAIC) of the training data. The WAIC score is a Bayesian adjusted metric to calculate the marginal likelihood (Watanabe, 2013). This metric has been shown by Choi & Jang (2018) to be robust towards the pitfall causing likelihood estimation models to assign high likelihoods to OoD data. The score is defined as follows:

$$\text{WAIC}(\boldsymbol{x}) = \mathbb{E}_{\alpha \sim p(\alpha)}[\log p_\alpha(\boldsymbol{x})] - \mathbb{V}_{\alpha \sim p(\alpha)}[\log p_\alpha(\boldsymbol{x})]. \tag{1}$$

Here, $\mathbb{E}[\cdot]$ and $\mathbb{V}[\cdot]$ denote expectation and variance respectively, which are taken over all architectures $\alpha$ sampled from the posterior architecture distribution $p(\alpha)$. Such a strategy captures model uncertainty in a Bayesian fashion, improving OoD detection. Intuitively, minimizing the variance of training data likelihoods allows its likelihood distribution to remain tight which, by proxy, minimizes the overlap of in-distribution and out-of-distribution likelihoods, thus making them separable.

Under this objective function, we search for an optimal distribution of network architectures $p(\alpha)$ by deriving the corresponding parameters that characterize $p(\alpha)$. Because the score requires aggregating the results from multiple architectures $\alpha$, optimizing such a score using existing search methods can be intractable, as they typically only consider a single architecture at a time. Later, we will show how to circumvent this problem in our optimization formulation.

## 3.2 SEARCH SPACE

NADS constructs a layer-wise search space with a pre-defined macro-architecture, where each layer can have a different architecture component. Such a search space has been studied by (Zoph & Le, 2016; Liu et al., 2018b; Real et al., 2019), where it shows to be both expressive and scalable/efficient.

The macro-architecture closely follows the Glow architecture presented in Kingma & Dhariwal (2018). Here, each layer consists of an actnorm, an invertible $1 \times 1$ convolution, and an affine coupling layer. Instead of pre-defining the affine coupling layer, we allow it to be optimized by our architecture search. The search space can be viewed in Figure 1. Here, each operational block of the affine coupling layer is selected from a list of candidate operations that include $3 \times 3$ average pooling, $3 \times 3$ max pooling, skip-connections, $3 \times 3$ and $5 \times 5$ separable convolutions, $3 \times 3$ and $5 \times 5$ dilated convolutions, identity, and zero. We choose this search space to answer the following questions towards better architectures for OoD detection:

- What topology of connections between layers is best for uncertainty quantification? Traditional likelihood estimation architectures focus only on feedforward connections without adding any skip-connection structures. However, adding skip-connections may improve optimization speed and stability.
- Are more features/filters better for OoD detection? More feature outputs of each layer should lead to a more expressive model. However, if many of those features are redundant, it may slow down learning, overfitting nuisances and resulting in sub-optimal models.
- Which operations are best for OoD detection? Intuitively, operations such as max/average pooling should not be preferred, as they discard information of the original data point "too aggressively". However, this intuition remains to be confirmed.

### 3.3 OPTIMIZATION

Having specified our proxy task and search space, we now describe our optimization method for NADS. Several difficulties arise when attempting to optimize this setup. First, optimizing $p(\alpha)$, a distribution over high-dimensional discrete random variables $\alpha$, jointly with the network parameters is intractable as, at worst, each network's optimal parameters would need to be individually identified. Second, even if we relax the discrete search space, the objective function involves computing an expectation and variance over all possible discrete architectures. To alleviate these problems, we first introduce a continuous relaxation for the discrete search space, allowing us to approximately optimize the discrete architectures through backpropagation and weight sharing between common architecture blocks. We then approximate the stochastic objective by using Monte Carlo samples to estimate the expectation and variance.

Specifically, let $\mathcal{A}$ denote our discrete architecture search space and $\alpha \in \mathcal{A}$ be an architecture in this space. Let $l_{\theta^*}(\alpha)$ be the loss function of architecture $\alpha$ with its parameters set to $\theta^*$ such that it satisfies $\theta^* = \arg\min_\theta l(\theta|\alpha)$ for some loss function $l(\cdot)$. We are interested in finding a distribution $p_\phi(\alpha)$ parameterized by $\phi$ that minimizes the expected loss of an architecture $\alpha$ sampled from it. We denote this loss function as $L(\phi) = \mathbb{E}_{\alpha \sim p_\phi(\alpha)}[l_{\theta^*}(\alpha)]$. For our NADS, this loss function is the negative WAIC score of in-distribution data $L(\phi) = -\sum_{i=1}^N \text{WAIC}(\boldsymbol{x}_i)$.

Solving $L(\phi)$ for arbitrary parameterizations of $p_\phi(\alpha)$ can be intractable, as the inner loss function $l_{\theta^*}(\alpha)$ involves searching for the optimal parameters $\theta^*$ of a neural network architecture $\alpha$. Moreover, the outer expectation causes backpropagation to be inapplicable due to the discrete random architecture variable $\alpha$. We adopt a tractable optimization paradigm to circumvent this problem through a specific reparameterization of the architecture distribution $p_\phi(\alpha)$, allowing us to backpropagate through the outer expectation and jointly optimize $\phi$ and $\theta$.

For clarity of exposition, we first focus on sampling an architecture with a single hidden layer. In this setting, we intend to find a probability vector $\phi = [\phi_1, \ldots, \phi_K]$ with which we randomly pick a single operation from a list of $K$ different operations $[o_1, \ldots, o_K]$. Let $\boldsymbol{b} = [b_1, \ldots, b_K]$ denote the random categorical indicator vector sampled from $\phi$, where $b_i$ is 1 if the $i^{th}$ operation is chosen, and zero otherwise. Note that $\boldsymbol{b}$ is equivalent to the discrete architecture variable $\alpha$ in this setting. With this, we can write the random output $\boldsymbol{y}$ of the hidden layer given input $\boldsymbol{x}$ as

$$\boldsymbol{y} = \sum_{i=1}^K b_i \cdot o_i(\boldsymbol{x}). \tag{2}$$

To make optimization tractable, we relax the discrete mask $\boldsymbol{b}$ to be a continuous random variable $\tilde{\boldsymbol{b}}$ using the Gumbel-Softmax reparameterization (Gumbel, 1954; Maddison et al., 2014) as follows:

$$\tilde{b}_i = \frac{\exp((\log(\phi_i) + g_i)/\tau)}{\sum_{j=1}^k \exp((\log(\phi_i) + g_i)/\tau)} \quad \text{for} \quad i = 1, \ldots, K. \tag{3}$$

Here, $g_1 \ldots g_k \sim -\log(-\log(u))$ where $u \sim \text{Unif}(0, 1)$, and $\tau$ is a temperature parameter. For low values of $\tau$, $\tilde{\boldsymbol{b}}$ approaches a sample of a categorical random variable, recovering the original discrete problem. While for high values, $\tilde{\boldsymbol{b}}$ will equally weigh the $K$ operations (Jang et al., 2016). Using this, we can compute backpropagation by approximating the gradient of the discrete architecture $\alpha$ with the gradient of the continuously relaxed categorical random variable $\tilde{\boldsymbol{b}}$, as $\nabla_{\theta,\phi}\alpha = \nabla_{\theta,\phi}\boldsymbol{b} \approx \nabla_{\theta,\phi}\tilde{\boldsymbol{b}}$. With this backpropagation gradient defined, generalizing the above setting to architectures with multiple layers simply involves recursively applying the above gradient relaxation to each layer.

With this formulation, we can gradually remove the continuous relaxation and sample discrete architectures by annealing the temperature parameter $\tau$. With this, we are able to optimize the architecture distribution $p_\phi(\alpha)$ and sample candidate architectures for further retraining, finetuning, or evaluation. By sampling $M$ architectures from the distribution, we are able to approximate the WAIC score expectation and variance terms as:

$$-L(\phi) = \sum_{i=1}^N \text{WAIC}(\boldsymbol{x}_i) \approx \sum_{i=1}^N \left[ \sum_{j=1}^M \log p_{\alpha_j}(\boldsymbol{x}_i) - \left( \sum_{j=1}^M (\log p_{\alpha_j}(\boldsymbol{x}_i))^2 - \left( \sum_{j=1}^M \log p_{\alpha_j}(\boldsymbol{x}_i) \right)^2 \right) \right]. \tag{4}$$

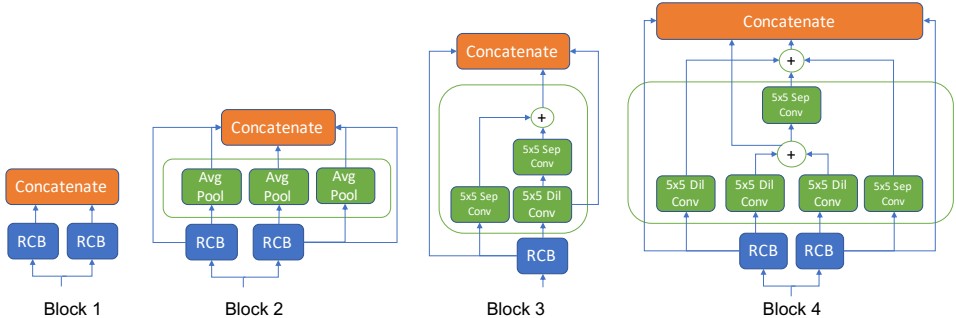

Figure 2: Summary of our architecture search findings: the most likely architecture structure for each block $K$ found by NADS.

## 3.4 SEARCH RESULTS

We applied our architecture search on five datasets: CelebA (Liu et al.), CIFAR-10, CIFAR-100, (Krizhevsky et al., 2009), SVHN (Netzer et al., 2011), and MNIST (LeCun). In all experiments, we used the Adam optimizer with a fixed learning rate of $1 \times 10^{-5}$ with a batch size of 4 for 10000 iterations. We approximate the WAIC score using $M = 4$ architecture samples, and set the temperature parameter $\tau = 1.5$ . The number of layers and latent dimensions is the same as in the original Glow architecture (Kingma & Dhariwal, 2018), with 4 blocks and 32 flows per block. Images were resized to $64 \times 64$ as inputs to the model. With this setup, we found that we are able to identify neural architectures in less than 1 GPU day.

Our findings are summarized in Figure 2, while more samples from our architecture search can be seen in Appendix C. Observing the most likely architecture components found on all of the datasets, a number of notable observations can be made:

- The first few layers have a simple feedforward structure, with either only a few convolutional operations or average pooling operations. On the other hand, more complicated structures with skip connections are preferred in the deeper layers of the network. We hypothesize that in the first few layers, simple feature extractors are sufficient to represent the data well. Indeed, recent work on analyzing neural networks for image data have shown that the first few layers have filters that are very similar to SIFT features or wavelet bases (Zeiler & Fergus, 2014; Lowe, 1999).

- The max pooling operation is almost never selected by the architecture search. This confirms our hypothesis that operations that discard information about the data is unsuitable for OoD detection. However, to our surprise, average pooling is preferred in the first layers of the network. We hypothesize that average pooling has a less severe effect in discarding information, as it can be thought of as a convolutional filter with uniform weights.

- The deeper layers prefer a more complicated structure, with some components recovering the skip connection structure of ResNets (He et al., 2016). We hypothesize that deeper layers may require more skip connections in order to feed a strong signal for the first few layers. This increases the speed and stability of training. Moreover, a larger number of features can be extracted using the more complicated architecture.

Interestingly enough, we found that the architectures that we sample from our NADS perform well in image generation without further retraining, as shown in Appendix D.

## 4 BAYESIAN MODEL ENSEMBLE OF NEURAL ARCHITECTURES

### 4.1 MODEL ENSEMBLE FORMULATION

Using the architectures sampled from our search, we create a Bayesian ensemble of models to estimate the WAIC score. Each model of our ensemble is weighted according to its probability, as in Hoeting et al. (1999). The log-likelihood estimate as well as the variance of this model ensemble

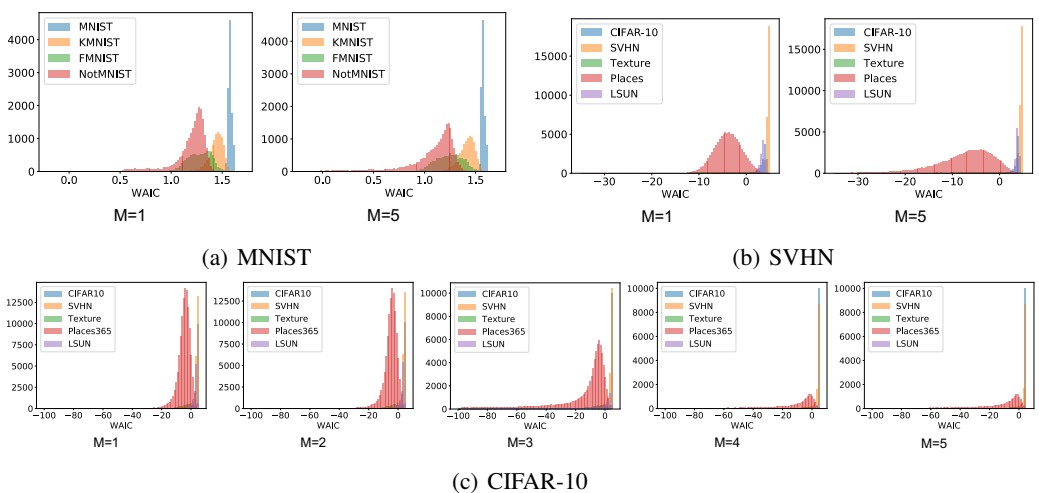

Figure 3: Effect of ensemble size to the distribution of WAIC scores estimated by model ensembles trained on different datasets. Larger ensemble sizes causes the WAIC score likelihood estimate of OoD data to be lower. Additional histograms for different ensemble sizes in Appendix F are with higher resolution.

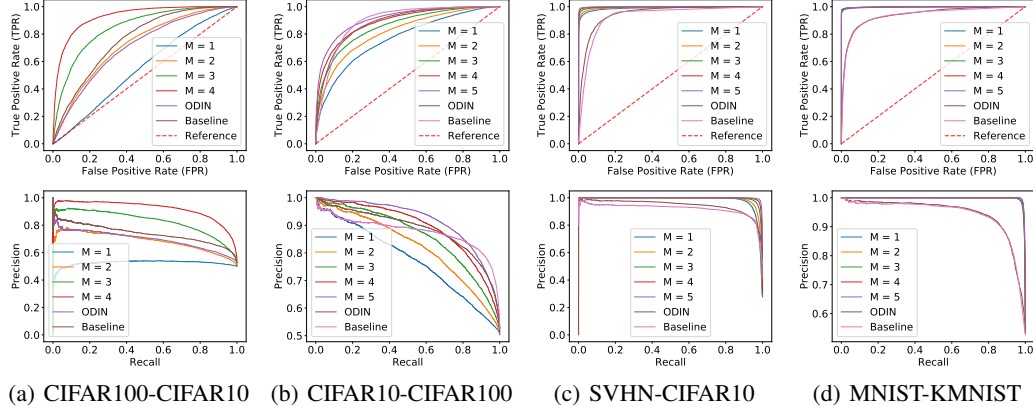

Figure 4: ROC and PR curve comparison of the most challenging evaluation setups for our NADS ensemble. Here, 'Baseline' denotes the method proposed by Hendrycks & Gimpel (2016). Subcaptions denote training-testing set pairs. Additional figures are provided in Appendix G.

is given as follows:

$$\mathbb{E}_{\alpha \sim p_\phi(\alpha)}[\log p(\boldsymbol{x})] = \sum_{\alpha \in \mathcal{A}} p_\phi(\alpha) \log p_\alpha(\boldsymbol{x}) \approx \sum_{i=1}^{M} \frac{p_\phi(\alpha_i)}{\sum_{j=1}^{M} p_\phi(\alpha_j)} \log p_{\alpha_i}(\boldsymbol{x}) \tag{5}$$

$$\mathbb{V}_{\alpha \sim p_\phi(\alpha)}[\log p(\boldsymbol{x})] \approx \sum_{i=1}^{M} \frac{p_\phi(\alpha_i)}{\sum_{j=1}^{M} p_\phi(\alpha_j)} \Big( \mathbb{V}[\log p_{\alpha_i}(\boldsymbol{x})] + (\log p_{\alpha_i}(\boldsymbol{x}))^2 \Big) - \mathbb{E}_{\alpha \sim p_\phi(\alpha)}[\log p(\boldsymbol{x})]^2 \tag{6}$$

Intuitively, we are weighing each member of the ensemble by their posterior architecture distribution $p_\phi(\alpha)$, a measure of how likely each architecture is in optimizing the WAIC score. We note that for our setup, $\mathbb{V}[\log p_{\alpha_i}(x)]$ is zero for each model in our ensemble; however, for models which do have variance estimates, such as models that incorporate variational dropout (Gal et al., 2017; Kingma et al., 2015; Gal & Ghahramani, 2016), this term may be nonzero. Using these estimates, we are able to approximate the WAIC score in equation (1).

## 4.2 ENSEMBLE RESULTS

We trained our proposed method on 4 datasets: CIFAR-10, CIFAR-100 (Krizhevsky et al., 2009), SVHN (Netzer et al., 2011), and MNIST (LeCun). In all experiments, we randomly sampled an

Table 1: OoD detection results on various training and testing experiments. We compared our method with MSP (Hendrycks & Gimpel, 2016), and Outlier Exposure (OE) (Hendrycks et al., 2019b).

| $D_{in}$ | $D_{out}$ | FPR% at TPR 95% | | | AUROC% | | | AUPR% | | |
|---|---|---|---|---|---|---|---|---|---|---|
| | | Base | OE | Ours | Base | OE | Ours | Base | OE | Ours |
| MNIST | not-MNIST | 10.3 | 0.25 | **0.00** | 97.2 | 99.86 | **100** | 97.4 | 99.86 | **100** |
| | F-MNIST | 61.1 | 0.99 | **0.00** | 88.8 | 99.83 | **100** | 90.8 | 99.83 | **100** |
| | K-MNIST | 29.6 | **0.03** | 0.76 | 93.6 | 97.60 | **99.80** | 94.3 | 97.05 | **99.84** |
| SVHN | Texture | 33.9 | 1.04 | **0.07** | 89.3 | **99.75** | 99.26 | 86.8 | **99.09** | 97.75 |
| | Places365 | 22.2 | 0.02 | **0.00** | 92.8 | **99.99** | 99.99 | 99.7 | **99.99** | 99.99 |
| | LSUN | 26.8 | 0.05 | **0.02** | 88.2 | 99.98 | **99.99** | 90.4 | 99.95 | **99.99** |
| | CIFAR10 | 23.2 | 3.11 | **0.37** | 91.1 | 99.26 | **99.92** | 91.9 | 97.88 | **99.83** |
| CIFAR10 | SVHN | 30.5 | **8.41** | 17.05 | 89.5 | **98.20** | 97.65 | 94.9 | 97.97 | **99.07** |
| | Texture | 39.8 | 14.9 | **0.25** | 87.7 | 96.7 | **99.81** | 79.8 | 94.39 | **99.86** |
| | Places365 | 36.0 | 19.07 | **0.00** | 88.1 | 95.41 | **100** | 99.5 | 95.32 | **100** |
| | LSUN | 14.6 | 15.20 | **0.44** | 95.4 | 96.43 | **99.83** | 96.1 | 96.01 | **99.89** |
| | CIFAR100 | 33.1 | **26.59** | 36.36 | 88.7 | **92.93** | 91.23 | 87.7 | **92.13** | 91.60 |
| | Gaussian | 6.3 | 0.7 | **0.00** | 97.7 | 99.6 | **100** | 93.6 | 94.3 | **100** |
| | Rademacher | 6.9 | 0.5 | **0.00** | 96.9 | 99.8 | **100** | 89.7 | 97.4 | **100** |
| CIFAR100 | SVHN | 46.2 | **42.9** | 45.92 | 82.7 | 86.9 | **94.35** | 91.3 | 80.21 | **96.01** |
| | Texture | 74.3 | 55.97 | **0.42** | 72.6 | 84.23 | **99.76** | 60.1 | 75.76 | **99.81** |
| | Places365 | 63.2 | 57.77 | **0.012** | 76.2 | 82.65 | **99.99** | 98.9 | 81.47 | **99.99** |
| | LSUN | 69.4 | 57.5 | **38.85** | 83.7 | 83.4 | **90.65** | 70.1 | 77.85 | **90.61** |
| | CIFAR10 | 62.5 | 59.96 | **34.41** | 75.8 | 77.53 | **92.83** | 74.0 | 72.82 | **91.93** |
| | Gaussian | 29.3 | 12.1 | **0.00** | 86.5 | 95.7 | **100** | 66.1 | 71.1 | **100** |
| | Rademacher | 59.4 | 17.1 | **0.00** | 51.7 | 93.0 | **100** | 32.7 | 56.9 | **100** |

ensemble of $M = 5$ models from the posterior architecture distribution $p_{\phi^*}(\alpha)$ found by NADS. Although these models can sufficiently perform image synthesis without retraining as shown in Appendix D, we observed that further retraining these architectures led to a significant improvement in OoD detection. Because of this, we retrained each architecture on data likelihood maximization for 150000 iterations using Adam with a learning rate of $1 \times 10^{-5}$.

We first show the effects of increasing the ensemble size in Figure 3 and Appendix F. Here, we can see that increasing the ensemble size causes the OoD WAIC scores to decrease as their corresponding histograms shift away from the training data WAIC scores, thus improving OoD detection performance. Next, we compare our ensemble search method against a traditional ensembling method that uses a single Glow architecture trained with multiple random initializations. As shown in Table 2, we find that our method is superior compared to the traditional ensembling method when compared on OoD detection using CIFAR-10 as the training distribution.

We then compared our NADS ensemble OoD detection method for screening out samples from datasets that the original model was not trained on. For SVHN, we used the Texture, Places, LSUN, and CIFAR-10 as the OoD dataset. For CIFAR-10 and CIFAR-100, we used the SVHN, Texture, Places, LSUN, CIFAR-100 (CIFAR-10 for CIFAR-100) datasets, as well as the Gaussian and Rademacher distributions as the OoD dataset. Finally, for MNIST, we used the not-MNIST, F-MNIST, and K-MNIST datasets. We compared our method against a baseline method that uses maximum softmax probability (MSP) (Hendrycks & Gimpel, 2016), as well as two popular OoD detection methods: ODIN (Liang et al., 2017) and Outlier Exposure (OE) (Hendrycks et al., 2019b). ODIN attempts to calibrate the uncertainty estimates of an existing model by reweighing its output softmax score using a temperature parameter and through random perturbations of the input data. For this, we use DenseNet as the base model as described in (Liang et al., 2017). On the other hand, OE models are trained to minimize a loss regularized by an outlier exposure loss term, a loss term that requires access to OoD samples.

As shown in Table 1 and Table 3, our method outperforms the baseline MSP and ODIN significantly while performing better or comparably with OE, which requires OoD data during training, albeit not from the testing distribution. We plot Receiver Operating Characteristic (ROC) and Precision-Recall (PR) curves in Figure 4 and Appendix G for more comprehensive comparison. In particular, our method consistently achieves high area under PR curve (AUPR%), showing that we are especially

capable of screening out OoD data in settings where their occurrence is rare. Such a feature is important in situations where anomalies are sparse, yet have disastrous consequences. Notably, ODIN underperforms in screening out many OoD datasets, despite being able to reach the original reported performance when testing on LSUN using a CIFAR10 trained model. This suggests that ODIN may not be stable for use on different anomalous distributions.

## 5 CONCLUSION

Unlike NAS for common learning tasks, specifying a model and an objective to optimize for uncertainty estimation and outlier detection is not straightforward. Moreover, using a single model may not be sufficient to accurately quantify uncertainty and successfully screen out OoD data. We developed a novel neural architecture distribution search (NADS) formulation to identify a random ensemble of architectures that perform well on a given task. Instead of seeking to maximize the likelihood of in-distribution data which may cause OoD samples to be mistakenly given a higher likelihood, we developed a search algorithm to optimize the WAIC score, a Bayesian adjusted estimation of the data entropy. Using this formulation, we have identified several key features that make up good uncertainty quantification architectures, namely a simple structure in the shallower layers, use of information preserving operations, and a larger, more expressive structure with skip connections for deeper layers to ensure optimization stability. Using the architecture distribution learned by NADS, we then constructed an ensemble of models to estimate the data entropy using the WAIC score. We demonstrated the superiority of our method to existing OoD detection methods and showed that our method has highly competitive performance without requiring access to OoD samples. Overall, NADS as a new uncertainty-aware architecture search strategy enables model uncertainty quantification that is critical for more robust and generalizable deep learning, a crucial step in safely applying deep learning to healthcare, autonomous driving, and disaster response.

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

## A    FIXED MODEL ABLATION STUDY

Table 2: OoD detection results on various training and testing experiments comparing our method with a baseline ensembling method that uses a fixed architecture trained multiple times with different random initializations.

| $D_{in}$ | $D_{out}$ | FPR% at TPR 95% | | AUROC% | | AUPR% | |
|---|---|---|---|---|---|---|---|
| | | Base Ensemble | Ours | Base Ensemble | Ours | Base Ensemble | Ours |
| CIFAR10 | SVHN | 50.07 | **17.05** | 93.48 | **97.65** | 95.98 | **99.07** |
| | Texture | 6.22 | **0.25** | 97.68 | **99.81** | 97.44 | **99.86** |
| | Places365 | 1.03 | **0.00** | 99.59 | **100** | 99.97 | **100** |
| | LSUN | 34.35 | **0.44** | 91.55 | **99.83** | 92.15 | **99.89** |
| | CIFAR100 | 65.13 | **36.36** | 78.44 | **91.23** | 79.44 | **91.60** |
| | Gaussian | **0.00** | **0.00** | **100** | **100** | **100** | **100** |
| | Rademacher | **0.00** | **0.00** | **100** | **100** | **100** | **100** |

## B    OOD DETECTION PERFORMANCE COMPARISON WITH ODIN

Table 3: OoD detection results on various training and testing experiments comparing our method with ODIN (Liang et al., 2017).

| $D_{in}$ | $D_{out}$ | FPR% at TPR 95% | | AUROC% | | AUPR% | |
|---|---|---|---|---|---|---|---|
| | | ODIN | Ours | ODIN | Ours | ODIN | Ours |
| MNIST | not-MNIST | 8.7 | **0.00** | 98.2 | **100** | 98.0 | **100** |
| | F-MNIST | 65 | **0.00** | 88.6 | **100** | 90.5 | **100** |
| | K-MNIST | 36.5 | **0.76** | 94.0 | **99.80** | 94.6 | **99.84** |
| SVHN | Texture | 33.9 | **0.07** | 92.4 | **99.26** | 88.2 | **97.75** |
| | Places365 | 22.2 | **0.00** | 94.9 | **99.99** | 99.8 | **99.99** |
| | LSUN | 26.8 | **0.02** | 93.5 | **99.99** | 93.1 | **99.99** |
| | CIFAR10 | 21.6 | **0.37** | 94.8 | **99.92** | 94.4 | **99.83** |
| CIFAR10 | SVHN | 36.5 | **17.05** | 89.7 | **97.65** | 95.6 | **99.07** |
| | Texture | 76.2 | **0.25** | 81.4 | **99.81** | 76.7 | **99.86** |
| | Places365 | 44.0 | **0.00** | 89.0 | **100** | 99.6 | **100** |
| | LSUN | 3.9 | **0.44** | 99.2 | **99.83** | 99.2 | **99.89** |
| | CIFAR100 | 45.4 | **36.36** | 88.3 | **91.23** | 88.5 | **91.60** |
| | Gaussian | 0.1 | **0.00** | **100** | **100** | 99.9 | **100** |
| | Rademacher | 0.3 | **0.00** | 99.9 | **100** | 99.8 | **100** |
| CIFAR100 | SVHN | **32.8** | 45.92 | 90.3 | **94.35** | 95.3 | **96.01** |
| | Texture | 78.9 | **0.42** | 75.7 | **99.76** | 64.5 | **99.81** |
| | Places365 | 63.3 | **0.012** | 79.0 | **99.99** | 99.1 | **99.99** |
| | LSUN | **17.6** | 38.85 | **96.8** | 90.65 | **96.5** | 90.61 |
| | CIFAR10 | 78.2 | **34.41** | 70.6 | **92.83** | 69.7 | **91.93** |
| | Gaussian | 1.3 | **0.00** | 99.5 | **100** | 97.8 | **100** |
| | Rademacher | 13.8 | **0.00** | 92.7 | **100** | 75.0 | **100** |

## C  ADDITIONAL SAMPLE ARCHITECTURES

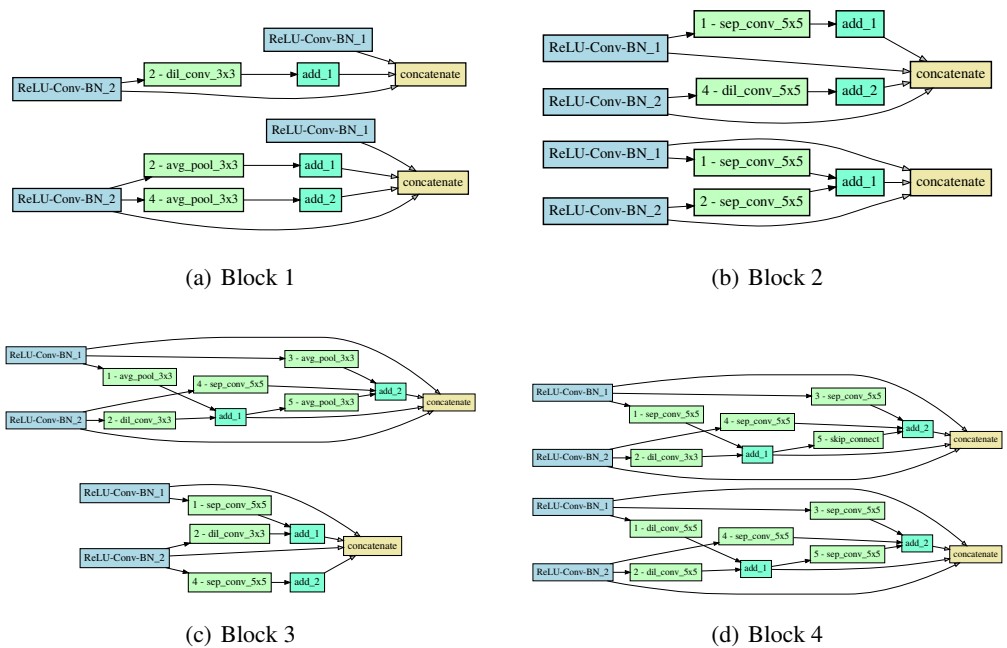

Figure 5: Maximum likelihood architectures inferred by our search algorithm on CelebA. Shown are two samples taken from each block.

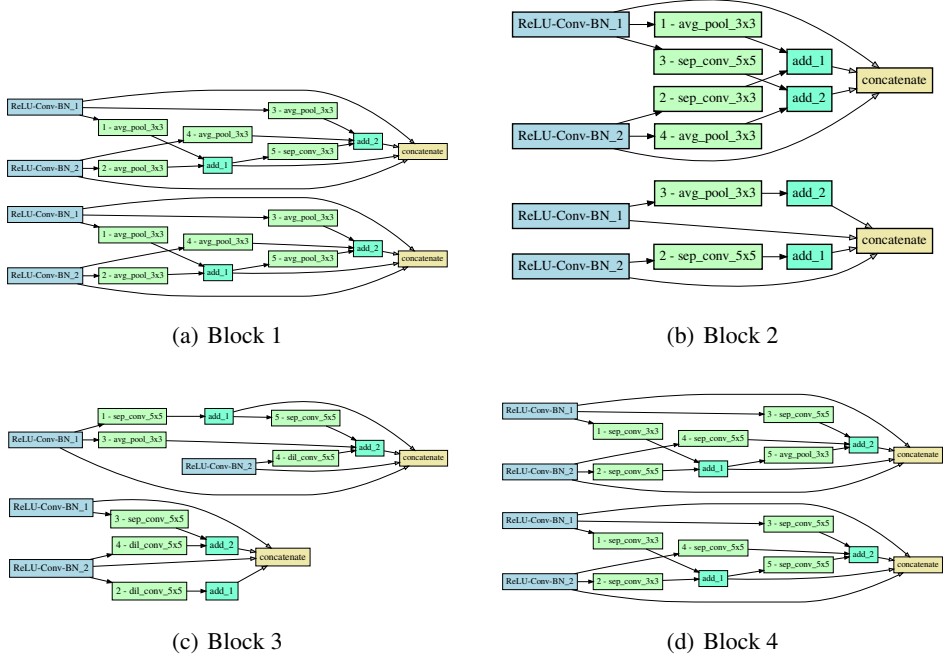

Figure 6: Maximum likelihood architectures inferred by our search algorithm on MNIST. Shown are two samples taken from each block.

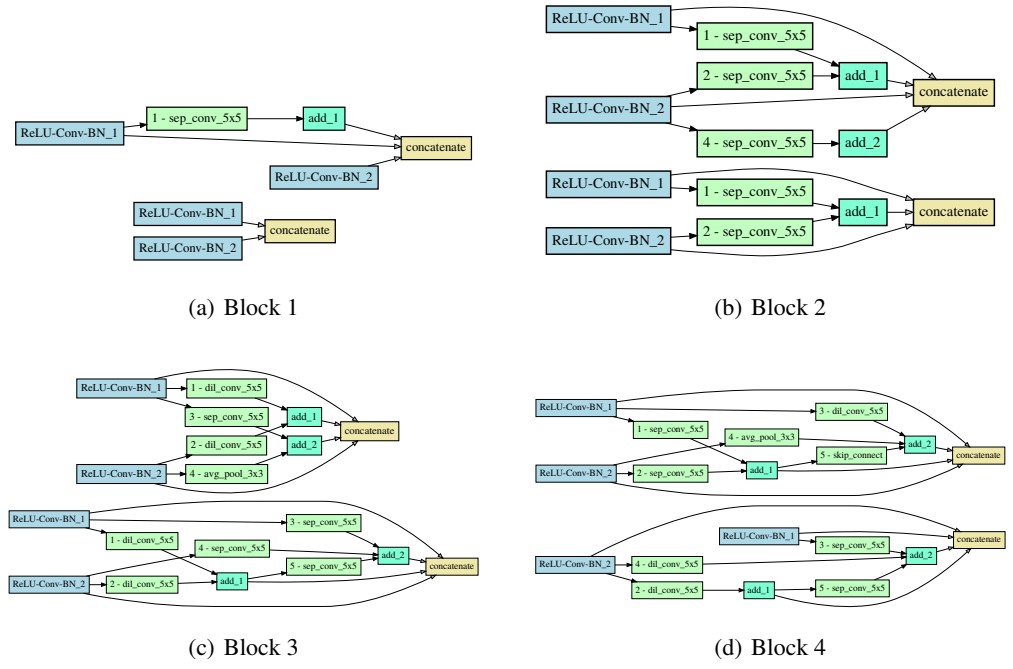

Figure 7: Maximum likelihood architectures inferred by our search algorithm on SVHN. Shown are two samples taken from each block.

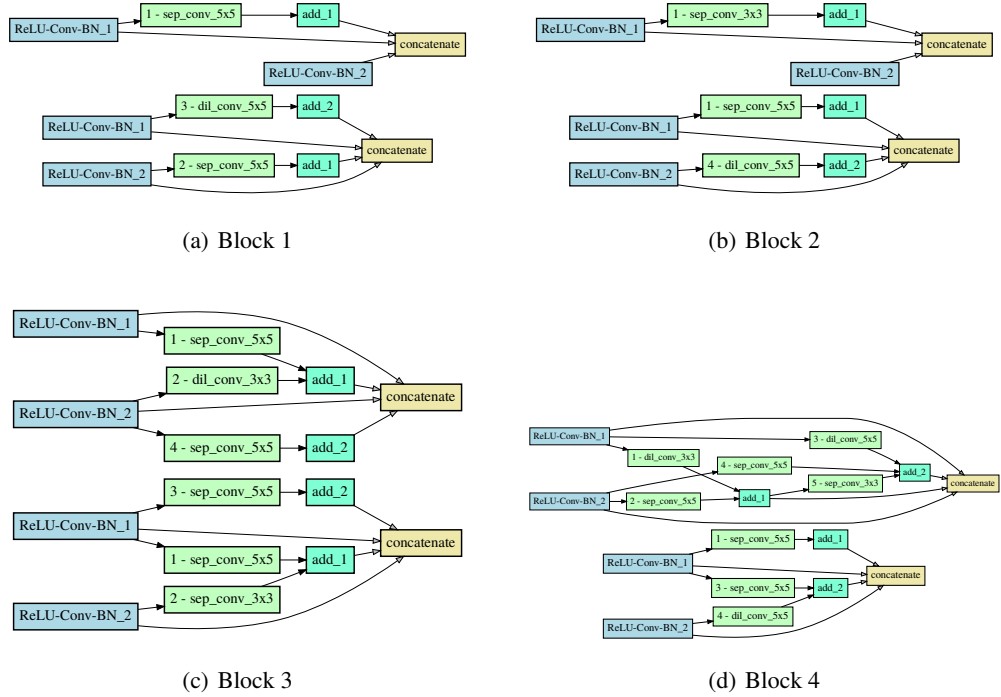

Figure 8: Maximum likelihood architectures inferred by our search algorithm on CIFAR-10. Shown are two samples taken from each block.

# D    IMAGE GENERATION SAMPLES

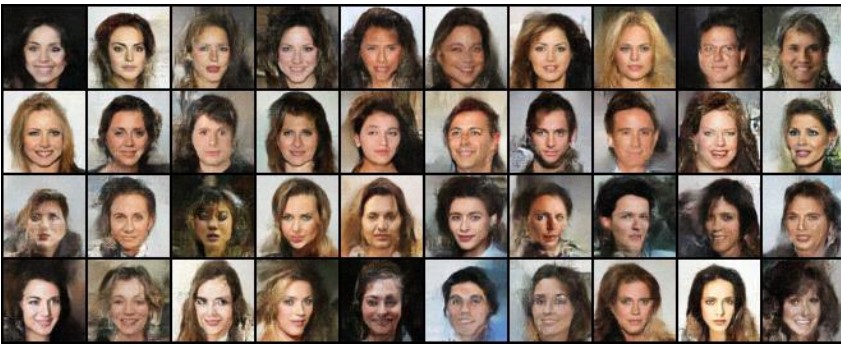

Figure 9: Samples taken from randomly sampled NADS architectures searched on CelebA. Images were not cherry-picked and the architectures were sampled without further retraining.

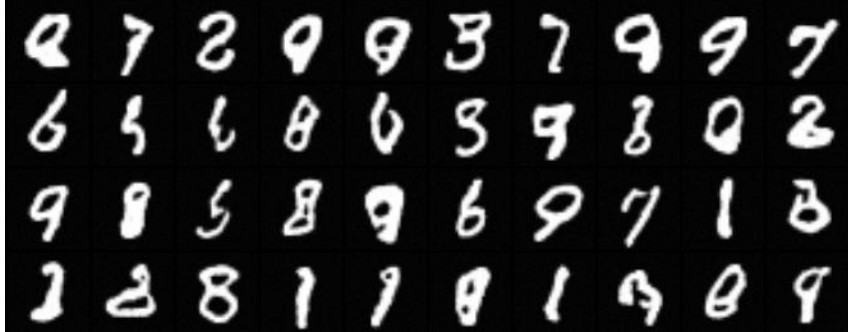

Figure 10: Samples taken from randomly sampled NADS architectures searched on MNIST. Images were not cherry-picked and the architectures were sampled without further retraining.

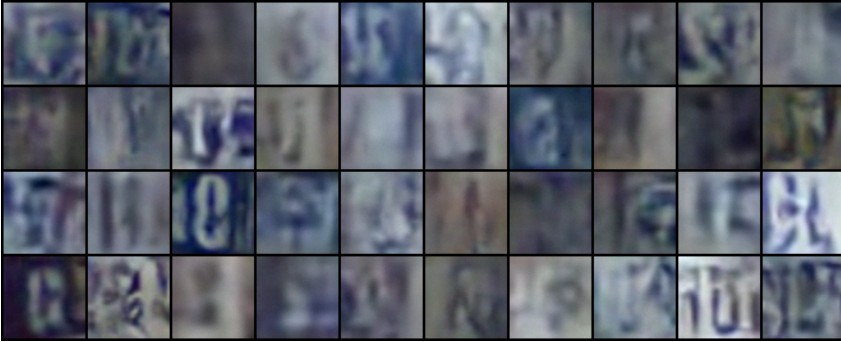

Figure 11: Samples taken from randomly sampled NADS architectures searched on SVHN. Images were not cherry-picked and the architectures were sampled without further retraining.

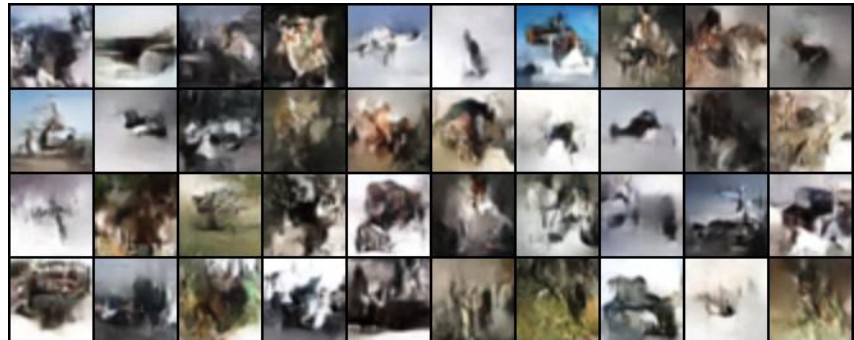

Figure 12: Samples taken from randomly sampled NADS architectures searched on CIFAR-10. Images were not cherry-picked and the architectures were sampled without further retraining.

# E  LIKELIHOOD ESTIMATION MODELS ASSIGN HIGHER LIKELIHOOD TO OOD DATA

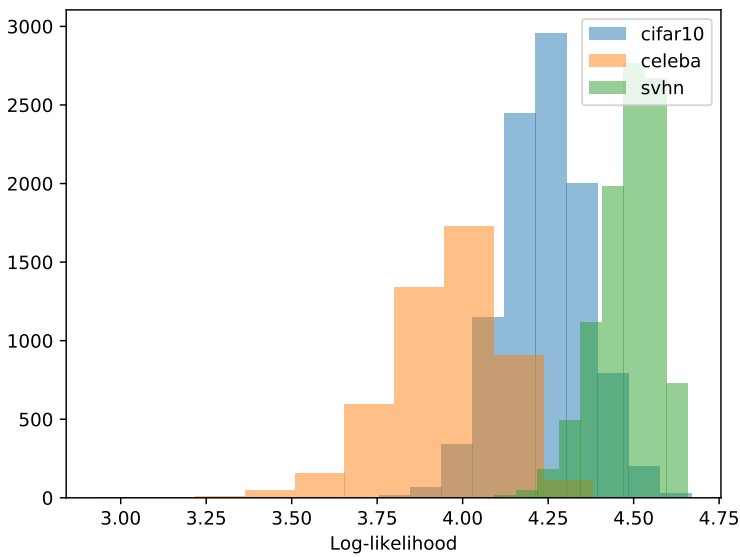

Figure 13: Likelihood distributions of different datasets evaluated on a Glow model trained on CelebA. The model assigns higher likelihood to OoD samples from CIFAR-10 and SVHN.

# F    EFFECT OF ENSEMBLE SIZE

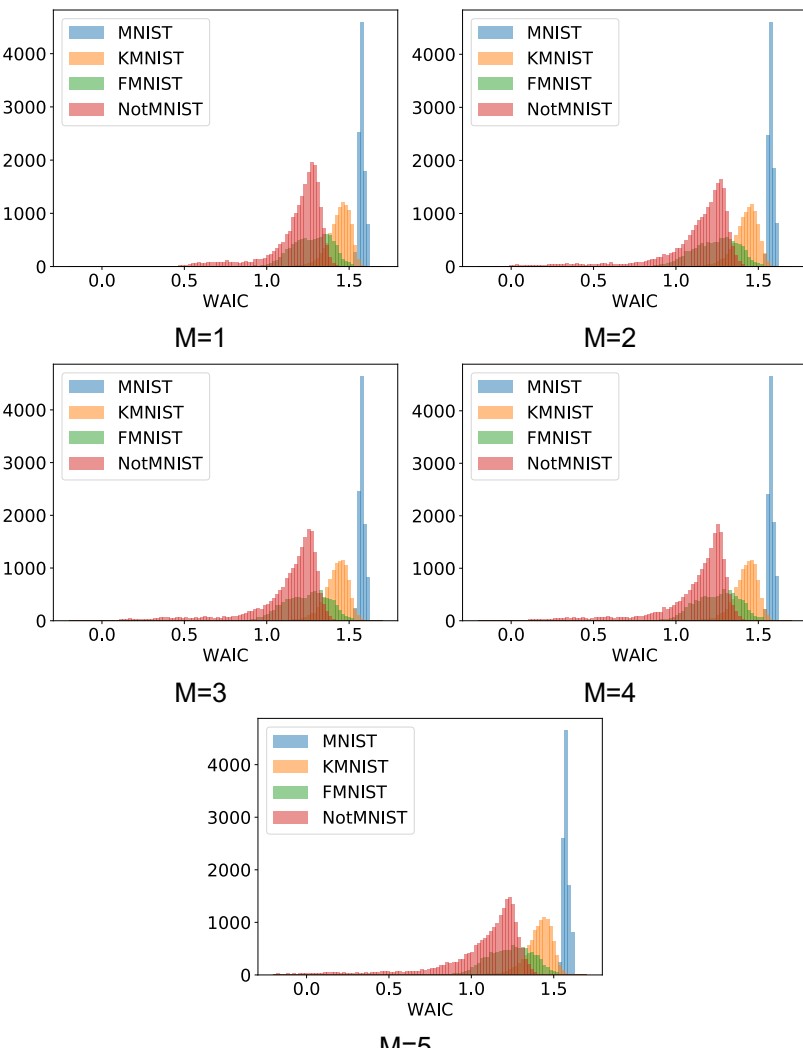

Figure 14: Effect of ensemble size to the distribution of WAIC scores estimated by model ensembles trained on MNIST.

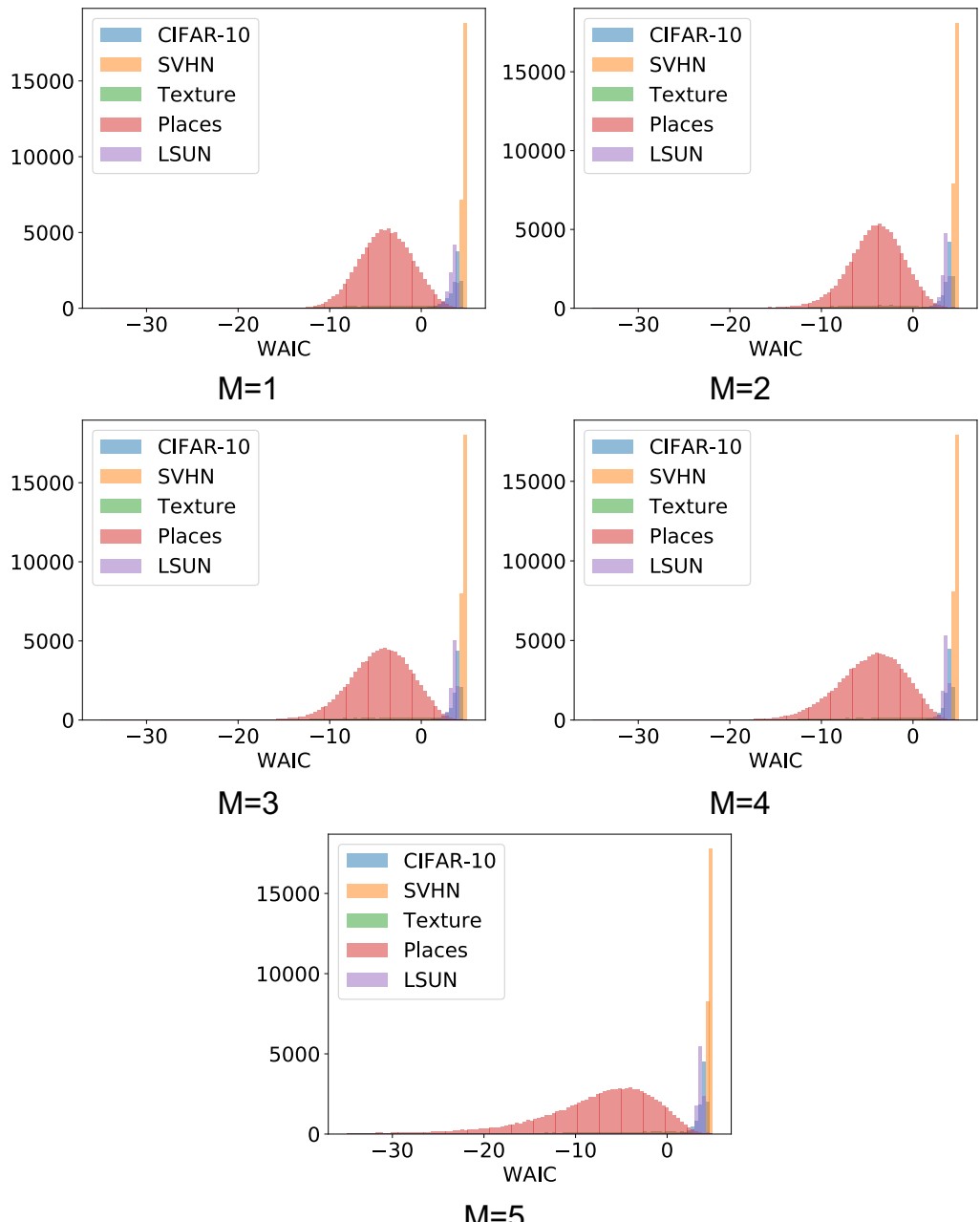

Figure 15: Effect of ensemble size to the distribution of WAIC scores estimated by model ensembles trained on SVHN.

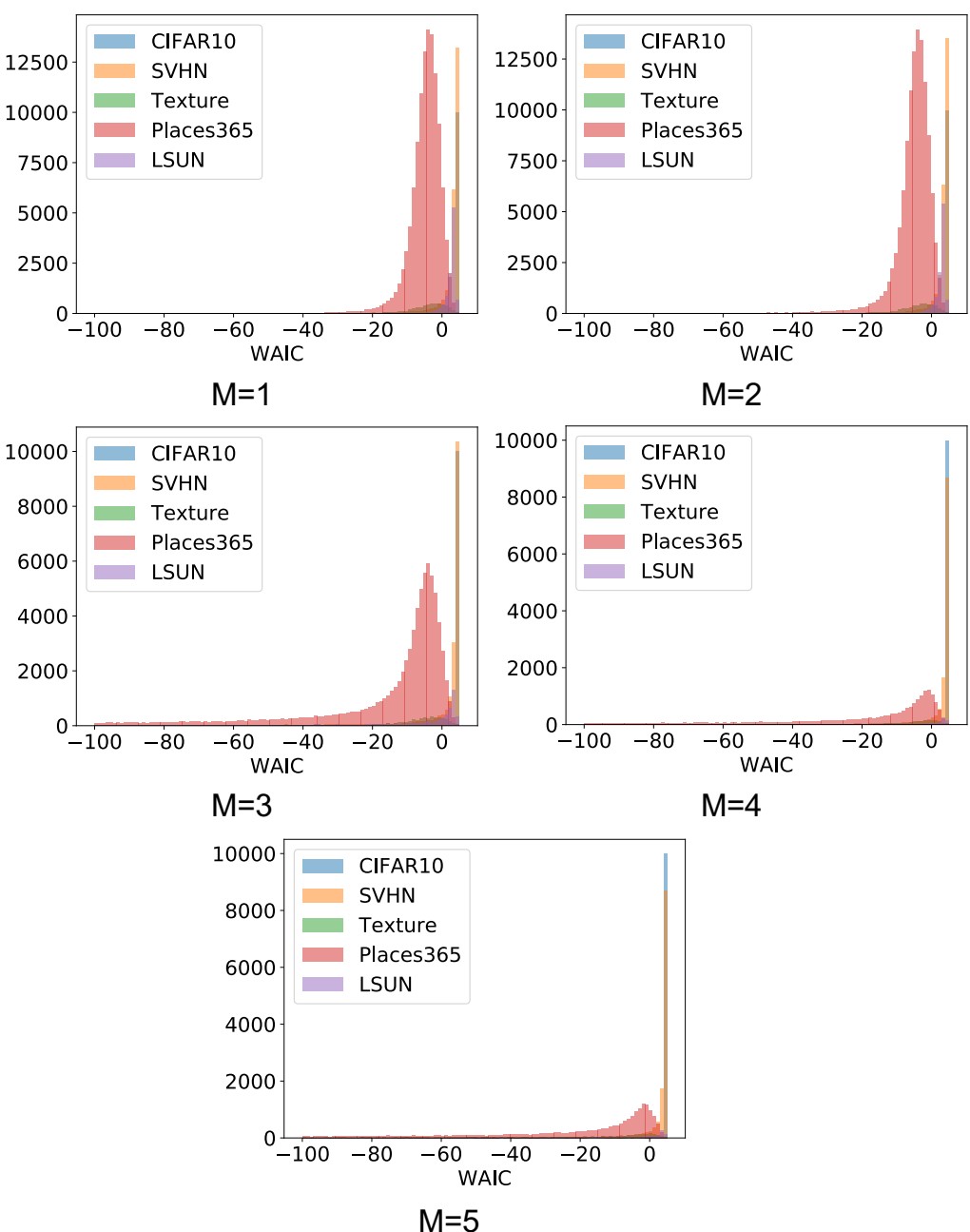

Figure 16: Effect of ensemble size to the distribution of WAIC scores estimated by model ensembles trained on CIFAR-10.

# G  ADDITIONAL ROC AND PRECISION-RECALL CURVES

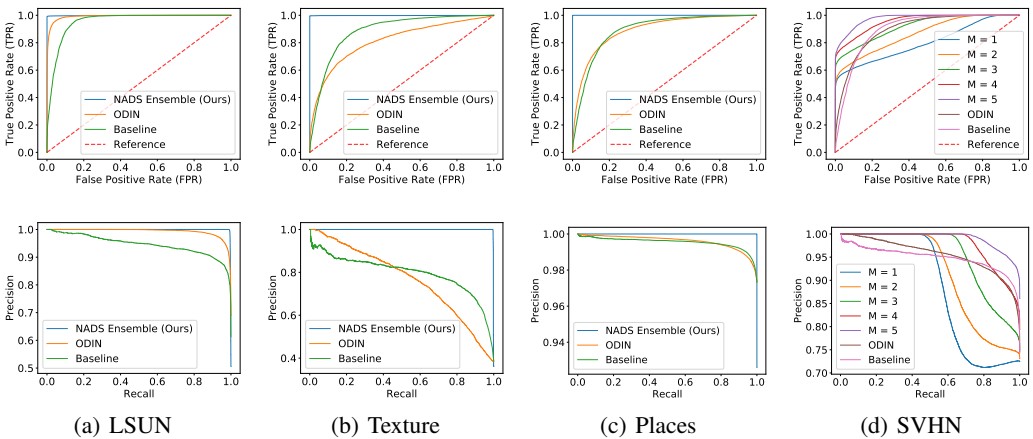

Figure 17: ROC and PR curve comparison of methods trained on CIFAR-10

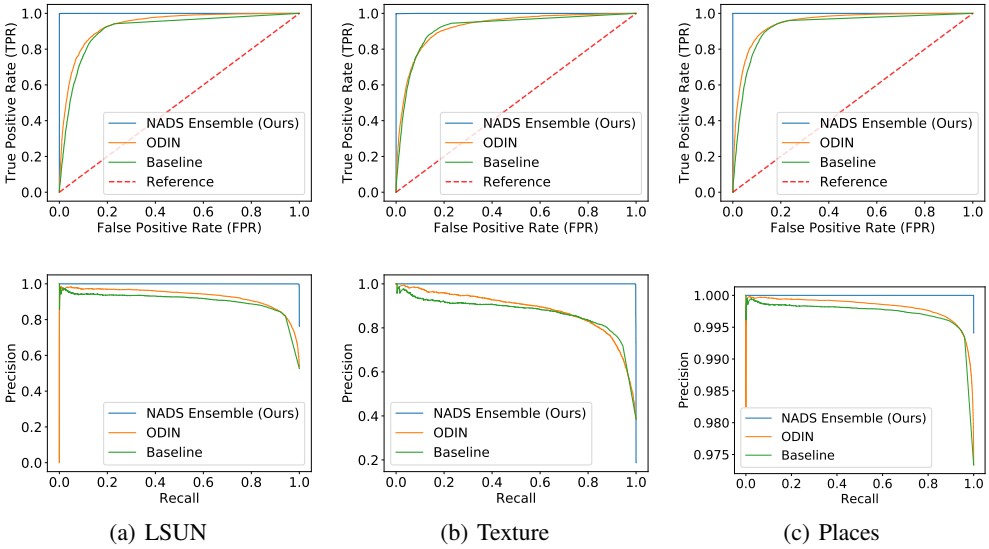

Figure 18: ROC and PR curve comparison of methods trained on SVHN

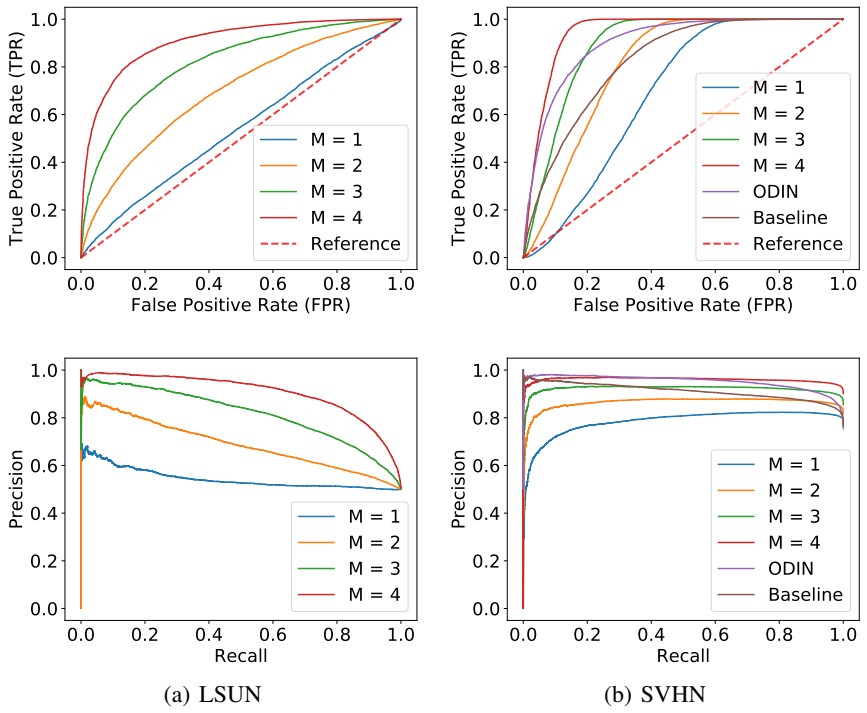

Figure 19: ROC and PR curve comparison of methods trained on CIFAR-100

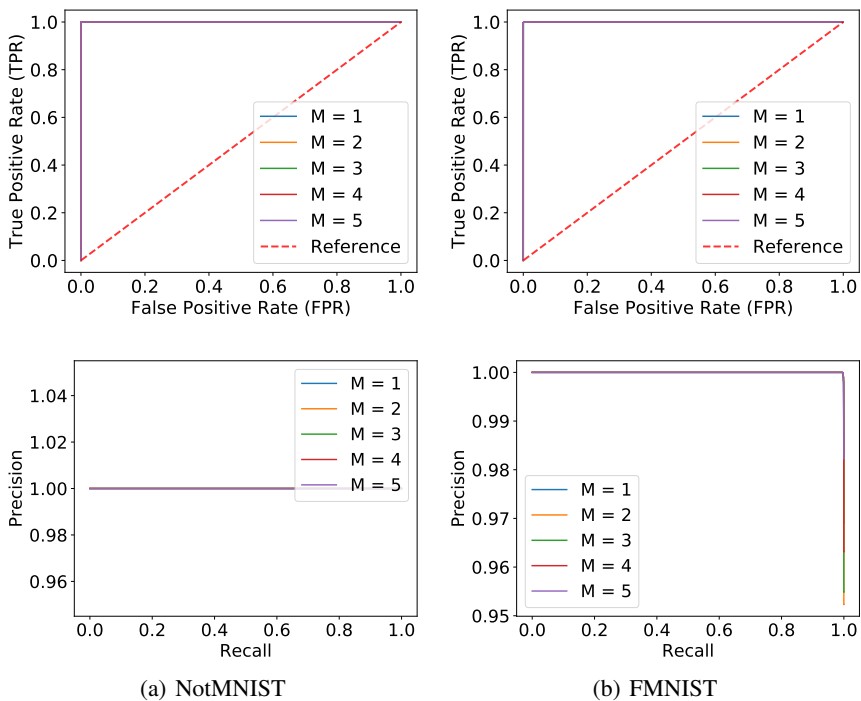

Figure 20: ROC and PR curve comparison of methods trained on MNIST

