# OpenReview forum: "NADS: Neural Architecture Distribution Search for Uncertainty Awareness"
_ICLR.cc/2020/Conference — Reject_

### Official Review · AnonReviewer3 · 2019-10-19
**Official Blind Review #3**

**Rating:** 8

**Review:**

This paper needs crucial, quick fixes.

This paper throws neural architecture search at the problem of out-of-distribution detection. Rather than searching over multi-class classifier architectures, and using the result for OOD detection, they instead search for generative model architectures.
The approach appears to work on some of their cherry-picked OOD datasets.

I give this paper a 3 because they are quite possibly only showing their strongest results and not giving a complete picture, and there are numerous small errors throughout the paper.
After a more thorough evaluation, the technique will likely not look strong on datasets such as CIFAR-10 vs CIFAR-100.
This is acceptable since they are comparing density estimators vs multi-class OOD detectors, the latter of which has been vastly superior for many years.
Their technique brings density estimators within striking distance of multi-class classifier perofrmance, but the paper must give a complete picture. If these issues are fixed, the paper is easily a 6 or an 8, depending on the results of currently unshown OOD datasets. It is acceptable if their technique gets and 55% AUROC for CIFAR-10 vs CIFAR-100 while multi-class classifiers with Outlier Exposure get 96%. It is OK because this technique appears superior to other generative models, which have lagged far behind. Currently the paper leaves the impression that this is not only leapfrogging past previous density estimators, but that it also is beating multi-class classifiers. This likely isn't true. We need to see performance on other OOD datasets.
I am willing to increase my score from a 6 to an 8 even if the results are negative.

The CIFAR-10 model should have to detect OOD samples from CIFAR-100, Rademacher/Bernoulli noise, and Gaussian noise. In addition, they should train a model on CIFAR-100 since many OOD detection techniques exhibit much worse behavior on CIFAR-100, compared to CIFAR-10 or MNIST or SVHN.

In summary, I give this a 3 due to critical flaws, but if these are rectified, the paper will likely deserve a 6 or 8.

Miscellaneous Points:

Please include CIFAR-10/100 code. Currently the code is for MNIST.

There are several errors in discussing related work.

> Moreover, previous work on deep uncertainty quantification shows that a single model may not suffice to quantify uncertainty and detect OoD samples (Lakshminarayanan et al., 2017; Choi & Jang, 2018)

Ensembles do not perform appreciably better at OOD detection under systematic OOD benchmarking when using multi-class classifiers. For generative models, ensembles can help. Ensembles mainly help multi-class classifier _calibration_ on in-distribution in-class data, but they have miniscule to nonexistent utility in classifier OOD detection. Please add appropriate qualifiers.

> "With no access to OoD data, unsupervised/self-supervised generative models which maximize the likelihood of in-distribution data become the primary tools for uncertainty quantification."
I think they mean "with access to labels." Multi-class classifiers are the most performant tool for OOD detection, and unsupervised generative models are around chance-levels. _Using Self-Supervised Learning Can Improve Model Robustness and Uncertainty_ (NeurIPS) uses self-supervised learning for OOD detection and achieves good performance, but this work is not cited.

> "However, these models counter-intuitively assign high likelihoods to OoD data (Nalisnick et al., 2019a; Choi & Jang, 2018)"
This was shown in previous work, such as Shafaei et al. 2019 and Hendrycks et al. 2019, and hence deserve mention.

> "Moreover, existing methods to calibrate model uncertainty estimates assume access to OoD data during training (Lee et al., 2018; Hendrycks et al., 2019). This is flawed when anomalous data is rare or not known ahead of time"
Lee et al. does not assume access to OOD data during training; they synthesize their own. Hendrycks et al. does assume access to OOD data, but not OOD data seen during evaluation; in this way, they do not assume data is "known ahead of time," which they reiterate throughout their paper. This sentence is painting with too broad a brush.

There is a smaller experimental problem. They compare to ODIN, but ODIN assumes access to OOD data from the test distribution. While this assumption is clearly questionable, their evaluation does not use the assumption they did, so their technique is not actually ODIN. I suggest just comparing against the Maximum Softmax Probability Baseline in Table 1 since the rest of this paper assumes OOD examples are not known ahead of time.

Update: I have changed my score to an 8.

**Experience Assessment:**

I have published in this field for several years.

**Review Assessment: Checking Correctness Of Derivations And Theory:**

N/A

**Review Assessment: Checking Correctness Of Experiments:**

I carefully checked the experiments.

**Review Assessment: Thoroughness In Paper Reading:**

I read the paper thoroughly.

---

> ### Author Response · Authors · 2019-11-13
> **Response to Reviewer 3**
>
> Thank you so much for taking time to review our paper. We do appreciate your recognition of the contributions of our NADS for OoD tasks with generative models.
>
> We would like to guarantee the reviewer that we did not “cherry-pick” the results to present. We presented the results in the original submission due to the consideration that the trends would be similar across datasets and that their presentation would be redundant. We have now included all the results together with the new experiments suggested by the reviewers.
>
> Following your suggestions, we have added additional testing configurations for our CIFAR-10 model, testing on Gaussian and Rademacher distributions generated the same way as was done on the Outlier Exposure (OE) paper, as well as on the CIFAR-100 dataset (page 8, Table 1). We have also trained a CIFAR-100 model and tested on configurations similar to their CIFAR-10 counterpart. Our results still show the competitive performance of our method. However, we agree that adding these additional testing configurations showed more failure cases of our proposed method.
>
> “Please include CIFAR-10/100 code. Currently the code is for MNIST.”:
>
> We have added this in the link. The code is the same as the one for MNIST but applied to a different dataset.
>
> "Moreover, previous work on deep uncertainty quantification shows that a single model may not suffice to quantify uncertainty and detect OoD samples":
>
> We have modified our manuscript to make this statement more specifically refer to generative models.
>
> "With no access to OoD data, unsupervised/self-supervised generative models which maximize the likelihood of in-distribution data become the primary tools for uncertainty quantification.":
>
> You are correct, this is what we meant by our statement. We have modified it in the revised manuscript to make this clearer.
>
> “Using Self-Supervised Learning Can Improve Model Robustness and Uncertainty_ (NeurIPS) uses self-supervised learning for OOD detection and achieves good performance, but this work is not cited.”
> "This was shown in previous work, such as Shafaei et al. 2019 and Hendrycks et al. 2019, and hence deserve mention.
> ":
>
> We have added the works above as additional citations in our manuscript.
>
> "but not OOD data seen during evaluation; in this way, they do not assume data is "known ahead of time,"":
>
> We feel that by assuming access to any OoD data, we are directly constraining the model's capacity from learning more unseen OoD data in future. Such an assumption can be incorrect when dealing with active/online learning situations where new training distributions are regularly encountered. However, we agree that the above statement slightly misrepresents current work on multi-class OoD classifiers. We have modified our wording to make it more appropriate.
>
> "There is a smaller experimental problem. They compare to ODIN, but ODIN assumes access to OOD data from the test distribution.":
>
> In our implementation of ODIN, we tuned with one OoD dataset 1000 samples. This is in line with what the ODIN authors suggested, as they state that the hyperparameters are not very sensitive to specific OoD datasets. We believe that this is an accurate implementation of ODIN. However, following your suggestion, we have added the MSP method as a baseline and moved our ODIN results to the appendix (page 13, Table 3) due to space constraints.

---

> > ### Comment · AnonReviewer3 · 2019-11-13
> > **Updated Score**
> >
> > The strong CIFAR-100 (in) CIFAR-10 (out) detection results have prompted me to update my score to an 8, as I indicated I would do in the original review.

---

### Official Review · AnonReviewer2 · 2019-10-21
**Official Blind Review #2**

**Rating:** 1

**Review:**

The authors propose a neural architecture search (NAS) method to construct a Bayesian ensemble of deep learning models. This ensemble is then employed to detect out-of-distribution examples.


The authors propose to use a differentiable architecture search method which model the architectural parameters using a concrete distribution. This idea was originally proposed by Xie et al. (2019) but this work was not discussed. Similarly, the work by Chang et al. (2019) is not discussed. In my opinion the novelty with respect to NAS is the WAIC objective function and its application to out-of-distribution detection.


The idea of using ensemble to detect out-of-distribution examples is not new. The authors already refer to the works by Choi & Jang (2018) and Lakshminarayanan et al. (2017). I'd like to add MC-Dropout (Gal et al., 2016) to this list which was used e.g. to detect adversarial examples.


The experimental section is well-written and the proposed method is able to outperform the chosen baselines. Obvious baselines are missing. There is no experiment that proof that this way of searching architectures finds better suited ensembles. How about maximizing the cross-entropy and train the discovered architecture multiple times from scratch and use these models in an ensemble to detect out-of-distribution examples? How about any ensemble-based method mentioned in the previous paragraph?


Concluding, the idea is nice but based on the current state of the paper it seems incremental. Experiments to back the usefulness of the described method are missing.


Sirui Xie, Hehui Zheng, Chunxiao Liu, Liang Lin: SNAS: stochastic neural architecture search. ICLR 2019
Jianlong Chang, Xinbang Zhang, Yiwen Guo, Gaofeng Meng, Shiming Xiang, Chunhong Pan: Differentiable Architecture Search with Ensemble Gumbel-Softmax. arXiv (2019)
Yarin Gal, Zoubin Ghahramani: Dropout as a Bayesian Approximation: Representing Model Uncertainty in Deep Learning. ICML 2016: 1050-1059

**Experience Assessment:**

I have published in this field for several years.

**Review Assessment: Checking Correctness Of Derivations And Theory:**

I assessed the sensibility of the derivations and theory.

**Review Assessment: Checking Correctness Of Experiments:**

I carefully checked the experiments.

**Review Assessment: Thoroughness In Paper Reading:**

I read the paper thoroughly.

---

> ### Author Response · Authors · 2019-11-13
> **Response to Reviewer 2**
>
> We thank the reviewer for the critiques, but have to emphasize the differences regarding the novelty and contributions of our presented work. We emphasize that the novelty of the presented work is the search for a distribution of architectures that are tailored to the tasks with Out-of-Distribution (OoD) testing examples. We also note that previous ensemble methods are not competitive against current OoD classification models such as Outlier Exposure (OE). To address concerns regarding the efficacy of our ensemble search method compared to using ensembles of a fixed model, we have included an additional ablation study (page 13, Table 2) comparing the two methods. Below, we address your questions and comments in detail.
>
> "This idea was originally proposed by Xie et al. (2019) but this work was not discussed. Similarly, the work by Chang et al. (2019) is not discussed.":
>
> We apologize that we did not discuss these two papers but we were not aware of these two previous works when we wrote the paper. We have updated our manuscript to include these citations. Although the relaxation method to the optimization problem is similar to ours in these papers, but they are still for maximum-likelihood-based neural architecture search. As we tried to emphasize in the original paper and response here to all the reviewers, compared to these two papers, the task, objective function, family of models, and our consideration to derive a Bayesian ensemble is novel. Specifically, to our knowledge, our formulation is the first work to apply NAS on likelihood estimation models, as well as on OoD detection. Our formulation is not supervised, as also noted by the third reviewer, compared to other NAS methods which mainly consider supervised classification models. Also, learning a distribution of architectures for generative models and using it for OoD detection has never been investigated before.
>
> "The idea of using ensemble to detect out-of-distribution examples is not new. The authors already refer to the works by Choi & Jang (2018) and Lakshminarayanan et al. (2017). I'd like to add MC-Dropout (Gal et al., 2016) to this list which was used e.g. to detect adversarial examples."
>
> We have tried the referred methods and they didn't perform competitively. For example, the likelihood score and WAIC score of MC-Dropout samples on OoD data were very similar to that of the training data. Comparing the results of Choi & Jang's (2018) work showed that their method is not competitive to ours on some testing configurations. To our knowledge, Outlier Exposure (OE), to which our method is compared against, is currently the most competitive method for OoD detection.
>
> "There is no experiment that proof that this way of searching architectures finds better suited ensembles."
>
> We did not compare against other ensemble methods because, to our knowledge, Outlier Exposure is currently the most competitive method for OoD detection. However, your concern is valid. To address this, we have included an additional ablation study in the appendix (page 13, Table 2) to show the effect of learning a distribution over the architecture space to better capture model architecture uncertainty. In this new experiment, we compared our proposed method against an ensemble of models with fixed architectures trained under different random initializations. The experiment clearly shows the improved performance of our proposed method.

---

### Official Review · AnonReviewer1 · 2019-10-22
**Official Blind Review #1**

**Rating:** 3

**Review:**

This paper considers the neural architecture search (NAS) problem under the out-of-distribution (OoD) environment. As the OoD problem is not visited in the current NAS literature, this paper proposes replacements for each of the three standard components in NAS, i.e., the proxy task, the search space, and the optimization algorithm; and each replacement is built upon an ensemble of existing techniques. Experiments are further verified on CelebA, CIFAR-10, SVHN, and MNIST.

Overall, the novelty in this paper is limited and experiments are not very convincing. Please see the questions below:

Q1. In the introduction, the authors write "Machine learning systems often encounter OoD errors when dealing with testing data coming from a different distribution from the one used for training".
- What is the validation set used in this paper?
- For the NAS problem, the architectural parameters must be guided by the validation set. So, does the validation set follows the same distribution as the training data set or the testing data set?
- If the validation set and the testing set have the same distribution, is it still a meaningful OoD problem?

Q2. "For example, naively using data likelihood maximization as a proxy task would run into the issue pointed out by Nalisnick et al. (2019a), with models assigning higher likelihoods to OoD data".
- How can I see this point from the given experiments?
- Is it better adding this into an ablation study?

Q3. Except for WAIC, what other metrics can we consider? The authors should have a more comprehensive related work section, which includes discussion on this part.
- Is it more meaningful to search a better metric than search architectures (which is just a standard applicaiton)?

Q4. In Section 3.4:  "CelebA (Liu et al.), CIFAR-10 (Krizhevsky et al., 2009), SVHN (Netzer et al., 2011), and MNIST (LeCun)" are used.
- Could the authors explain more tasks and details on these data sets? Specifically, why they are OoD problems.
- Based on the descriptions from the authors, these data sets seem to be standard ones.

Q5. Variation is already considered in (1), I mean the second term there.
- Why should we still consider an ensemble of models?
- Is it better adding an ablation study on M about " ROC and PR curve"? (not Figure 13, 14 in the appendix)

Q6. Is it better to have a comparison with standard NAS in the experiments?
- While authors argue they are not applicable here, it is still good to demonstrate how not applicable they are.
- If the validation set follows the same distribution as the testing set, gradient signals on architectural parameters perhaps can still be helpful.

Q7. Can natural gradient descent be applied to (4)?
- "First, optimizing p(α), a probability over ..., each network’s optimal parameters would need to be individually". The first problem is not a really challenging problem please have a check at "Adaptive Stochastic Natural Gradient Method for One-Shot Neural Architecture Search".

Q8. What is the searching time of the proposed method? How's it compared with recent NAS methods? e.g. DARTS (DARTS: Differentiable Architecture Search).

**Experience Assessment:**

I have published one or two papers in this area.

**Review Assessment: Checking Correctness Of Derivations And Theory:**

I assessed the sensibility of the derivations and theory.

**Review Assessment: Checking Correctness Of Experiments:**

I assessed the sensibility of the experiments.

**Review Assessment: Thoroughness In Paper Reading:**

I read the paper at least twice and used my best judgement in assessing the paper.

---

> ### Author Response · Authors · 2019-11-13
> **Response to Reviewer 1**
>
> Thank you very much for your review of our paper. We emphasize that the novelty of the presented work is the search for a distribution of architectures that are tailored to the tasks with Out-of-Distribution (OoD) testing examples. To the best of our knowledge, existing neural architecture search (NAS) methods only optimize for maximum likelihood problems with point estimates, which is not appropriate in the OoD environment. Below please find our response to your questions/comments:
>
> Q1:
> As we search for a neural architecture distribution instead of tuning specific hyperparameters, we did not use any validation set for model selection in our work. With our method, we are able to optimize the neural architecture without using a validation set. However, it is true that other authors have used a validation set for NAS to prevent overfitting. Our NADS can be considered as a Bayesian version of previous NAS methods. It naturally enables uncertainty quantification and alleviates potential overfitting issues. We note again that the training procedure of our NADS does not see any testing data.
>
> Q2:
> This issue has been discussed by several previously published papers, such as the one you cited. When we wrote the paper, we did not think that this issue needed further restatement. The issue can be seen in Figure 3(c), where some samples of SVHN are incorrectly given higher likelihood than CIFAR10 for M=1, the original training data. We have modified our manuscript to include an additional figure in the appendix to show this issue more clearly. More comprehensive discussions of the problem can be found in the following papers:
>
> Choi, Hyunsun, Eric Jang, and Alexander A. Alemi. "Waic, but why? generative ensembles for robust anomaly detection." arXiv preprint arXiv:1810.01392 (2018).
>
> Hendrycks, Dan, et al. "Using self-supervised learning can improve model robustness and uncertainty." arXiv preprint arXiv:1906.12340 (2019).
>
> Shafaei, Alireza, Mark Schmidt, and James J. Little. "A Less Biased Evaluation of Out-of-distribution Sample Detectors."
>
> Q3:
> As we are investigating likelihood-based models, other metrics that can be considered would be other forms of penalized likelihood. WAIC has been shown to be robust to the issue of assigning high likelihoods to OoD data, as likelihoods of OoD data have been shown to have higher variance compared with that of the training data. WAIC is uncertainty-aware for training Bayesian ensembles and WAIC-based training leads to more robust performance with respect to potential uncertainty. Hence, it is the most appropriate metric for our NADS training. This was actually observed empirically as we have tried other penalized likelihood methods that computes the likelihood through adversarial perturbations, and they do not perform as well as WAIC-based NADS does.
>
> Q4:
> As stated in our manuscript, we trained a likelihood estimation model on these 4 datasets. The likelihood estimation model is then used to discriminate OoD samples from other datasets. For example, the model trained on MNIST was used to discriminate MNIST from the OoD datasets K-MNIST, F-MNIST, and NotMNIST. Again, our NADS only sees the training dataset and the other datasets are hence considered as OoD data.
>
> Q5:
> - It is true that variation is already considered in (1). However, (1) was estimated by sampling. In the search stage, this was done by sampling multiple architectures. In the evaluation stage, the sampling was done by considering an ensemble of architectures, each architecture giving a sample to estimate (1).
> - As our NADS aims to search for a distribution of architectures that can better estimate the estimation uncertainty, the ensemble of models can be considered the empirical estimates from NADS.
> - We have added additional ROC and PR curves (Figure 4, page 7) to our manuscript to more clearly show the effect of the ensemble size on the model's performance.
>
> Q6:
> Standard NAS algorithms do not search for a distribution of architectures. As such, they are not able to optimize the WAIC as it needs to be approximated through sampling, or further approximated to make optimization tractable. Moreover, as we are not constructing any validation set from OoD data, they should not perform well in OoD detection, as they will likely run into the problem of assigning high likelihood to OoD data. We believe that searching for architecture distributions is indeed one of the unique contributions of our NADS.
>
> Q7:
> We agree on this. Our statement in the manuscript is that the NAS problem is intractable in general. Various relaxations to the original problem can be done to make it more computationally feasible, such as the Gumbel-Softmax reparameterization we used, or using the method referred above. We will revise our manuscript to make this statement clearer.
>
> Q8:
> Using our setup, we find that we are able to find good performing architectures in less than 1 GPU day. We have modified our manuscript to include this detail.

---

### Public Comment · ~Xuefeng_Du1 · 2021-01-30
**Questions about a claim in this ICLR submission and the final ICML paper**

Dear authors,

Sorry for bothering, but you state in the paper "We can gradually remove the continuous relaxation and sample discrete architectures by annealing the temperature parameter τ , allowing us to perform architecture search without using a validation set."

But it is not clear to me why "We can gradually remove the continuous relaxation and sample discrete architectures by annealing the temperature parameter τ , " leads to "allowing us to perform architecture search without using a validation set." Therefore, you are optimizing the arch parameters on the training set? It violates the setting of NAS or further hyperparameter optimization.

Could you help me understand your claim?

Thanks,

---

> ### Author Response · Authors · 2021-02-02
> **Response**
>
> Hi Xuefeng,
> Thank you for your interest in our paper! We address your question below:
>
> We optimize the distribution of architectures via the continuous relaxation using only the training data without requiring any validation data (an important consideration for OoD setups). In that sentence, we didn't mean specifically that annealing the temperature "leads to" allowing us to not use a validation set. Instead, we meant that by utilizing the continuous relaxation and optimizing the distribution of architectures, even with only the training data without any validation set (the entire discussion stated before the sentence regarding annealing), we can sample discrete architectures by annealing the temperature to achieve reliable predictions and uncertainty estimates.
>
> I hope this makes our discussion clearer.

---

### Decision · Program_Chairs · 2019-12-19

**Decision:**

Reject

**Comment:**

This paper introduces a neural architecture search method that is geared towards yielding good uncertainty estimates for out-of-distribution (OOD) samples.

The reviewers found that the OOD prediction results are strong, but criticized various points, including the presentation of the OOD results, novelty as a NAS paper, missing citations to some recent papers, and a lack of baselines with simpler ensembles.
The authors improved the presentation of their OOD results and provided new experiments, which causes one reviewer to increase his/her score from a weak reject to an accept. The other reviewers appreciated the rebuttal, but preferred not to change their scores from a weak reject and a reject, mostly due to lack of novelty as a NAS paper.

I also read the paper, and my personal opinion is that it would definitely be very novel to have a good neural architecture search for handling uncertainty in deep learning; it is by no means the case that "NAS for X" is not interesting just because there are now a few papers for "NAS for Y". As long as X is relevant (which uncertainty in deep learning definitely is), and NAS finds a new state-of-the-art, I think this is great. For such an "application" paper of the NAS methodology, I do not find it necessary to introduce a novel NAS method, but just applying an existing one would be fine. The problem is more that the paper claims to introduce a new method, but that that method is too similar to existing ones, without a comparison; actually just using an existing NAS method would therefore make the contribution and the emphasis on the application domain clearer.
I have one small question to the authors about a part that I did not understand: to optimize WAIC (Eq 1), why is it not optimal to just set the parameterization \phi such that the variance is minimized, i.e., return a delta distribution p_\phi that always returns the same architecture (one with a strong prediction)? Surely, that's not what the authors want, but wouldn't that minimize WAIC? I hope the authors will clarify this in a future version.

In the private discussion of reviewers and AC, the most positive reviewer emphasized that the OOD results are strong, but admitted that the mixed sentiment is understandable since people who do not follow OOD detection could miss the importance and context of the results, and that the paper could definitely improve its messaging. The other reviewers' scores remained at 1 and 3, but the reviewers indicated that they would be positive about a future version of the paper that fixed the identified issues. My recommendation is to reject the paper and encourage the authors to continue this work and resubmit an improved version to a future venue.